# Evaluation of the Physico-Chemical Properties of Liposomes Assembled from Bioconjugates of Anisic Acid with Phosphatidylcholine

**DOI:** 10.3390/ijms222313146

**Published:** 2021-12-05

**Authors:** Hanna Pruchnik, Anna Gliszczyńska, Aleksandra Włoch

**Affiliations:** 1Department of Physics and Biophysics, Wrocław University of Environmental and Life Sciences, Norwida 25, 50-375 Wrocław, Poland; hanna.pruchnik@upwr.edu.pl; 2Department of Chemistry, Wrocław University of Environmental and Life Sciences, Norwida 25, 50-375 Wrocław, Poland; anna.gliszczynska@upwr.edu.pl

**Keywords:** bioconjugates, liposomes, DPPC, biophysical methods, phase transition, fluidity, DSC, infrared spectroscopy

## Abstract

The aim of this work was the evaluation of the physico-chemical properties of a new type of liposomes that are composed of DPPC and bioconjugates of anisic acid with phosphatidylcholine. In particular, the impact of modified anisic acid phospholipids on the thermotropic parameters of liposomes was determined, which is crucial for using them as potential carriers of active substances in cancer therapies. Their properties were determined using three biophysical methods, namely differential scanning calorimetry (DSC), steady-state fluorimetry and attenuated total reflectance Fourier transform infrared spectroscopy (ATR-FTIR). Moreover, temperature studies of liposomes composed of DPPC and bioconjugates of anisic acid with phosphatidylcholine provided information about the phase transition, fluidity regarding chain order, hydration and dynamics. The DSC results show that the main phase transition peak for conjugates of anisic acid with phosphatidylcholine molecules was broadened and shifted to a lower temperature in a concentration- and structure-dependent manner. The ATR-FTIR results and the results of measurements conducted using fluorescent probes located at different regions in the lipid bilayer are in line with DSC. The results show that the new bioconjugates with phosphatidylcholine have a significant impact on the physico-chemical properties of a membrane and cause a decrease in the temperature of the main phase transition. The consequence of this is greater fluidity of the lipid bilayer.

## 1. Introduction

Liposomes have been extensively investigated as effective drug delivery systems (DDS) since 1965, when they were introduced for the first time by Alec Bangham et al. [1]. Their unique properties, including biocompatibility and biodegradability, as well as their physiochemical nature, make them excellent vesicles for both hydrophilic and hydrophobic molecules [2]. They are a kind of soft material, easily bottom-up, self-organized and self-assembled to unilamellar vesicles, which are small enough to venture into the finest capillaries. In the early 1970s, the potential of liposomes as drug delivery vehicles in antitumor therapy was underlined for the first time by Gregoriadis and co-workers [3]. However, in the following years, their numerous disadvantages were also reported. The main issues they present are prematurely enacting the burst release of encapsulated active molecules and low loading efficacy. Therefore, research on liposome technology progressed during the following decades from conventional vesicles (“first-generation liposomes”) to “second-generation liposomes” which were obtained by modulating the lipid composition, size and charge of the vesicles. Moreover, this strategy, based on the modification of a chemical surface of liposomes by incorporating specific molecules or macromolecules, such as glycolipids or sialic acid, was also presented as a method for enhancing their interaction with a target site [4]. A significant step in the development of long-circulating liposomes came with the inclusion of the synthetic polymer poly-(ethylene glycol) (PEG) in liposome composition. In the early 1990s, many liposomal formulations, including Doxil^®^, Ambisome^®^, DepoDur^®^, Myocet^®^ and DepoCyt^®^, were approved for clinical practice [5,6]. Many others are also currently under clinical trials, but the main problems they present are still their ineffectiveness in releasing encapsulated drugs and finding a way to open up efficient transport routes into the target cells.

Recently, a novel concept of self-assembled nanocarriers that can be applied for drug delivery has been reported in the literature [7,8]. These new nanocarriers are based on the novel biomaterials that are obtained by the conjugation of active molecules/drugs with phospholipids by replacing one or two hydrophobic fatty acids chains of phosphatidylcholine (PC) with hydrophobic or hydrophilic drugs/active molecules. This type of bioconjugate is able to self-assemble and forms nanostructures which are, in turn, effective nanocarriers for active molecule delivery. It was proven that liposomes formed from phospholipid bioconjugates have superior drug-loading abilities and avoid premature burst releases in comparison with conventional drug-loaded nanocarriers. The research group of Li demonstrated how liposomal formulation derived from a dual-chlorambucil-tailed phospholipid was developed to deliver drugs and significantly improve antitumor efficacy in a way that reduced side effects, a fact that highlights their potential applications in combinatorial therapy [9].

In our previous paper, we reported phospholipid conjugates with biologically active molecules [10,11,12,13,14], and their high antiproliferative activity towards selected cancer cell lines was proved. Moreover, for these conjugates, it was also confirmed that at the established active concentrations, they are characterized by the selectivity of actions against neoplastic cells, remaining non-toxic to cells with an undisturbed proliferation process. Therefore, in the current study, the conjugates of phosphatidylcholine with *p*-anisic acid were applied to form liposomes in combination with major membrane lipids—DPPC. The encapsulation of insoluble compounds into liposomes obviates the necessity for a solvent. The properties of liposomes in aqueous environments have to be described to enable their potential applications in liposome systems as a drug or as drug carriers. Whereas the physico-chemical properties of a lipid bilayer are important parameters in designing and producing lipid-based drug delivery systems [15,16], the aim of this work was to study the impact of two newly synthesized conjugates phosphatidylcholines containing anisic acid at the *sn*-1 and *sn*-2 positions on the physico-chemical properties of liposomes (Figure 1).

In particular, the effect on the thermotropic parameters of liposomes was determined using three biophysical methods, including: differential scanning calorimetry (DSC), steady-state fluorimetry and attenuated total reflectance Fourier transform infrared spectroscopy (ATR-FTIR). Additionally, the influence of anisic acid on the DPPC phase transition was tested (Figure 1). Moreover, understanding the influence of these conjugates on the physico-chemical properties of membranes is crucial for their use as potential carriers of active substances in cancer therapies.

## 2. Results and Discussion

### 2.1. Differential Scanning Calorimetry (DSC)

The multilamellar liposomes, composed of DPPC, 1-ANISA-2-PA-PC/DPPC, 1-PA-2-ANISA-PC/DPPC and ANISA/DPPC at the molar ratios 1:5; 1:10; 1:25 and 1:50, were examined using differential scanning calorimetry (DSC). DSC is a fundamental technique used for obtaining a precise determination of thermodynamic parameters during structural changes or phase transitions in lipid bilayers. Thermodynamic parameters, which characterize interactions between the conjugates and DPPC, such as the temperature of the main phase transition (T_m_) and pretransition (T_p_), the half-width of the main peak (∆T_1/2_) as well as a change in enthalpy of main transition (∆H), were obtained. The impact of two conjugates, namely phosphatidylcholines containing anisic acid at the *sn*-1 and *sn*-2 positions and anisic acid on the phase transition of DPPC, is shown in Figure 2. According to literature data, for liposomes formed only from DPPC, two characteristic phase transitions have been observed, namely pretransition and the main phase transition, at temperatures of 35.4 °C (T_p_) and 41.8 °C (T_m_), respectively (Figure 2a–c). The pretransition takes place from the lamellar gel phase (L_β__’_) to the ripple phase transition (P_β__′_), whereas the main phase transition takes place from the ripple phase (P_β__′_) to the fluid lamellar phase transition (L_α_). In the temperature range T_p_–T_m_, an intermediate phase has been observed in which bilayers are modulated by a ripple phase.

The DSC measurements of liposome bilayers with different molar ratio conjugates showed that the temperature and enthalpy of the main phase transition of DPPC and pretransition are changed. The 1-PA-2-ANISA-PC compound slightly increased the half-width peak and decreased the enthalpy of the main phase transition, while the intermediate phase (P_β__′_) did not disappear. Similar changes in ∆H, T_m_ and ∆T_1/2_ were caused by the presence of anisic acid, except that at higher concentrations of the compound, for a molar ratio of ANISA/DPPC 1:10, a pretransition peak was not detected. Interestingly, the ripple phase for the 1-ANISA-2-PA-PC/DPPC system disappeared at a molar ratio of 1:25, and the peak half-width increased significantly from ∆T_1/2_ = 0.7 ± 0.2 °C for pure DPPC to ∆T_1/2_ = 3.3 ± 0.2 °C, and T_m_ decreased by about 1.8 °C compared to the control. Further increases in the content of 1-ANISA-2-PA-PC and 1-PA-2-ANISA-PC in liposomes formed from DPPC led to a decrease in the cooperativity and temperature of the main phase transition (Appendix A). At the molar ratio conjugates to DPPC of 1:1, the main phase transition was still visible, but the intermediate phase disappeared completely, probably indicating an increase in the lipid bilayer fluidity. At the molar ratio of 1:1, the differences between the newly synthesized conjugates studied were even more pronounced; the presence of 1-ANISA-2-PA-PC resulted in a significant blurring of the main phase transition peak; for the 1-ANISA-2-PA-PC/DPPC system, T_m_ = 37.3 ± 0.2 °C, and for 1-PA-2-ANISA-PC/DPPC system, T_m_ is 40.0 °C. The enthalpy value also decreased significantly, about 50% compared to the control. The presence of compounds in liposomes above a molar ratio of 1:10 caused an increase in the half-width of the main phase transition peak, especially for 1-ANISA-2-PA-PC. It is a well-known fact that there is a link between ∆T_1/2_ and both the cooperativity of a transition and the cooperativity of the main transition of DPPC bilayer, i.e., the former is inversely related to the latter. This is a key point considering how fast or how far the results of a trans-gauche conformational change will propagate within the membrane when one constituent phospholipid molecule underwent the conformational change [17]. Thus, the width of the main peak is related to the cooperativity of the phase transition, and the increasing presence of 1-ANISA-2-PA-PC and 1-PA-2-ANISA-PC significantly decreased the cooperativity.

The differences between the effects of 1-ANISA-2-PA-PC and 1-PA-2-ANISA-PC on ∆T_1/2_ as well as on T_m_ and T_p_ are most likely due to a difference in the conformation of those molecules with acid attachment at the *sn*-1 or *sn*-2 positions to phosphatidylocholine.

Our research is in line with that of other authors. For example, Arouri and Mouritsen [18] determined, using the DSC method, the main phase transition temperature of the retinoid-lipid prodrugs mixed with DPPC in various proportions of this compound (10–50 mol%). The authors noticed that with increasing mol% of the prodrug, followed by the widening of the DSC peaks of the mixtures, decreases in the transition temperature were observed. This suggests high perturbs in the acyl chain packing of DPPC in a concentration-dependent manner.

### 2.2. Spectroscopy Methods

In order to extend the research on the influence of newly synthesized conjugates on the thermotropic parameters of liposomes, spectroscopic methods, namely steady-state fluorescence and ATR-FTIR, were also used.

#### 2.2.1. Steady-State Fluorescence Spectroscopy

In order to obtain additional information regarding the influence of conjugates (1-ANISA-2-PA-PC, 1-PA-2-ANISA-PC and ANISA) on the physico-chemical properties of the liposome’s membrane, the fluorimetric method was also employed. In this research the conjugates/DPPC at the molar ratio of 1:5 were used. In these studies, three different fluorescent probes, namely MC540, Laurdan and DPH, were used in order to obtain information from three different areas of the membrane.

The first of them is the MC540 probe. It is a heterocyclic chromophore with a negative charge that binds to the outer leaflet of the phospholipid bilayer above the glycerol skeleton [19,20]. The presence of a negative charge may play an important role in the process of MC540 incorporation into the membrane [21]. MC540 is used to monitor the molecular packing of phospholipids in the hydrophilic part of the membrane [20]. These changes are monitored on the basis of the fluorescence intensity of the MC540 probe, whose maximum was recorded at 585 nm. Figure 3 shows the fluorescence intensity of MC540 for the control liposomes (DPPC) and for the new forms of liposomes (1-ANISA-2-PA-PC/DPPC, 1-PA-2-ANISA-PC/DPPC and ANISA/DPPC) as a function of temperature. The results show that the fluorescence intensity of MC540 for the control strongly increases in the gel phase and decreases in the liquid phase. These results are in line with previous studies described in the literature [22,23], which show that the fluorescence of merocyanine is temperature- and lipid fluidity-dependent. As was shown by Williamson and coauthors, the higher fluorescence intensity of MC540 is enhanced in the presence of disordered or fluid lipid membranes [22]. However, instead of a steplike increase in fluorescence, the maximum at the phase transition was observed. Experiments carried out by Langner and Hui show that the fluorescence intensity maximum at the main phase transition of lipid is based on both the change of membrane fluidity at the main phase transition and the superposition of a temperature effect on the probe′s quantum efficiency, which produces the apparent fluorescence maximum [23]. In the case of the new forms of liposomes, higher intensity of MC540 compared to the control in both the gel and fluid phases was observed. The greatest intensity compared to the control was in the case of liposomes composed from conjugates with *p*-anisic acid at the *sn*-1 position (1-ANISA-2-PA-PC/DPPC). Moreover, adding this compound led to the almost complete abolition of the main phase transition. The presence of *p*-anisic acid at *sn*-2 position or only anisic acid caused a shift in the temperature of the main phase transition of lipid (towards lower temperatures and a reduction in the size of peak). As was suggested by Langner and Hui [23], the increased fluorescence of MC540 in the presence of vesicles above their main phase transition indicates a fluid bilayer. The enhanced fluorescence of MC 540 suggests a decrease in the organization of lipids [22] and/or an increase in the membrane surface area available for the binding of the dye.

The second fluorescent probe, i.e., Laurdan, is located in the hydrophilic–hydrophobic interface of the bilayer. This probe is incorporated into the hydrophobic region of the membrane due to a 12-carbon aliphatic chain, but fluorophore is located at the level of the ester groups of lipids. It is extremely sensitive to polarity changes in the environment, which is related to the phospholipid phase state. These changes are reflected by shifts in the Laurdan emission spectrum in the gel and liquid crystalline phase. The maximum emission band in the gel phase occurs at 440 nm, then the lipids are slightly hydrated, and solvent relaxation does not take place. In the liquid crystalline phase, the maximum emission shifts to 490 nm, and the bilayer relaxes, which makes it easier for water molecules to penetrate at the level of the glycerol backbone [24,25]. Polarity changes were monitored by the generalized polarization parameter (GP). The values of GP for control liposomes (DPPC) and for new forms of liposomes as a function of temperature are shown in Figure 4a. The results show a significant decrease in the value of GP in all the new liposomes compared to the control in both the gel and liquid phase. The conjugates 1-ANISA-2-PA-PC and 1-PA-2-ANISA-PC also caused a shift in the temperature of the main phase transition of lipids towards lower temperatures compared to the control. Although the presence of *p*-anisic acid at the *sn*-2 or *sn*-1 position caused a more significant decrease in the temperature of the main phase transition than would have occurred if only anisic acid had been used, significant differences between acid attachment at *sn*-1 or *sn*-2 position were not observed. A decrease in the values of GP suggests a decrease in packing order in the hydrophilic–hydrophobic interface of the bilayer. This fact indicates that the conjugates of liposomes with *p*-anisic acid caused an increasing disorder in the polar heads of the lipid bilayer.

DPH is a hydrophobic probe that has an affinity for the hydrocarbon chains in the membrane. It is used to determine changes in fluidity and the main phase transition of lipids based on fluorescence anisotropy. Changes in anisotropy are connected with the mobility of the DPH probe in the hydrocarbon chains. When the hydrophobic area of a membrane is more ordered, anisotropy increases. A decrease in this value indicates an increase in the fluidity of the membrane (an increase in the mobility of the DPH probe in the membrane is associated with an increase in the mobility of the alkyl chains lipids). Figure 4b shows the values of the anisotropy of DPH for control liposomes (DPPC) and for liposomes composed from the conjugates 1-ANISA-2-PA-PC, 1-PA-2-ANISA-PC or ANISA and DPPC at the molar ratio of 1:5, as a function of temperature.

The results indicate that the conjugates of 1-ANISA-2-PA-PC and 1-PA-2-ANISA-PC had the value of anisotropy in the gel (up to 35 °C) and liquid phase (after main phase transition up to 50 °C) at the control level. However, a significant shift in the temperature of the main phase transition of lipids towards lower temperatures was observed for new forms of liposomes compared to the control. A greater effect was shown by the 1-ANISA-2-PA-PC conjugate than by 1-PA-2-ANISA-PC. In the case of ANISA, only an increase in anisotropy in the gel phase was seen. Interestingly, no major shifts in the temperature of the main phase transition were observed. Incorporating a compound in the hydrophobic region caused a decrease in the anisotropy parameter due to structural perturbation of the bilayer [26,27].

The results obtained using the fluorimetric method are consistent with those obtained using the DSC method. The results indicate that the new liposomes composed from conjugates of 1-ANISA-2-PA-PC and 1-PA-2-ANISA-PC have a significant impact on the physico-chemical properties of the membrane and cause a slow transition or are even able to abolish the lipid phase transition. The molecules, 1-ANISA-2-PA-PC in particular, cause the lipid bilayer to show greater fluidity. ANISA also has an impact on the physico-chemical properties of the membrane in the hydrophilic part and hydrophilic–hydrophobic interface of the bilayer. Its impact on the hydrophobic area is slight.

#### 2.2.2. Attenuated Total Reflectance Fourier Transform Infrared Spectroscopy (ATR-FTIR)

In order to understand the mechanism of molecular interaction between the 1-ANISA-2-PA-PC and 1-PA-2-ANISA-PC molecules and the DPPC bilayer, the ATR-FTIR technique was applied. Furthermore, in order to check the exact effects on the structure and phase transition of the new forms of conjugates and DPPC at the molar ratio of 1:5, measurements were performed over a wide range of temperatures, below and above the main phase transition of DPPC. IR measurement was performed for both the dehydrated DPPC film and the mixture of DPPC with conjugates of 1-ANISA-2-PA-PC and 1-PA-2-ANISA-PC film and hydrated liposomes composed of bioconjugates of anisic acid with phosphatidylcholine (1-ANISA-2-PA-PC/DPPC and 1-PA-2-ANISA-PC/DPPC). The most significant frequencies for all dehydrated probes are shown in Appendix A. The reviewed absorption spectra of dehydrated DPPC film and of dehydrated conjugates/DPPC film are shown in Figure 5a, and the selected bands’ spectra of hydrated DPPC, 1-ANISA-2-PA-PC/DPPC and 1-PA-2-ANISA-PC/DPPC systems are shown in Figure 5b. The shape and position of the maximum of the absorption band of the hydrated lipid bilayer are temperature-dependent [28]. It is highly likely that changes observed here were caused not only by increasing temperature, but also by a movement of different/additional molecules in the DPPC bilayer [29]. The interaction between 1-ANISA-2-PA-PC or 1-PA-2-ANISA-PC and DPPC molecules is manifested in the changes of the spectral parameters of hydrocarbon chains, the interfacial group and the polar head group. The shape and position of the analyzed bands (stretching vibration of ν_as_(CH_2_), ν_s_(CH_2_), ν(C=O), ν_as_(PO_2_^−^), ν_s_(PO_2_^−^)) changed with rising temperature (Figure 5b, Figure 6a,b and Figure 7a,b). The most significant frequencies of conjugates of liposomes with *p*-anisic acid and blank DPPC liposome in different temperatures (different phases) are shown in Table 1.

The region with the most intense vibrational spectra (3000–2900 cm^−1^) contains several stretching vibrations of C-H groups from phospholipid hydrocarbon chains. Within the spectrum of DPPC liposomes, one CH_2_ asymmetric stretching was located at about 2920 cm^−1^ and another—a symmetric one—at about 2850 cm^−1^ (Figure 5a,b). It is worth noting that both the frequencies and widths of these bands were vulnerable to conformational changes of lipid chains. This conformational fluidization of the DPPC bilayer may be explained by increasing temperature, water content or incorporating compounds into the lipid membrane. The increase in the wavenumber of these bands testifies to an increased fluidity of the hydrophobic part of the lipid bilayer. They respond to any differences that occur in the *trans/gauche* ratio in acyl chains.

A comparison of the infrared spectra of DPPC and the conjugates/DPPC in the region of the hydrophobic bilayer of lipid chains is presented in Figure 5b. Moreover, in Figure 6a,b full attention is given to illustrate a dependence of the asymmetric and symmetric CH_2_ stretching vibration in pure DPPC and the conjugates/DPPC systems as a function of temperature. The frequency of the methylene symmetric stretching vibrational mode is vulnerable to changes in the conformational order of lipid acyl chains; therefore, it seems to be useful in order to monitor the progress of the lipid-gel-to-liquid-phase transition (L_α_) [29,30]. Whereas the main phase transition is extremely sharp for the pure DPPC, for liposomes consisting of conjugates with *p*-anisic acid, in particular for 1-ANISA-2-PA-PC/DPPC, it becomes slightly wider and shifts towards lower temperatures. This suggests that some of the chains are in the *gauche* conformation, which may be due to their greater mobility. The fluidity of lipid bilayer in the presence of tested compounds increases, as evidenced by the values of the wavenumbers at specific temperatures.

Spectral modes arising from the head group and interfacial region of a lipid are also useful as valuable sources of information. One possible way of examining the structure of the interfacial region of lipid bilayer is via ester group vibration. From this point of view, the C=O stretching frequencies between bands 1750−1700 cm^−1^ are the most intense. In the dehydration samples two absorption bands are associated with two ester group in diacyl lipids. This splitting arises, at least partially, due to the process of conformational inequivalence about C_1_-C_2_ bonds’ *sn*-1 and *sn*-2 chains, which adopt *trans* and *gauche* conformations, as well as due to possible differences in the extent of hydration. As a consequence, the position of the ν(C=O) band’s maximum is dependent on the conformation of ester groups and to the hydration level of the carbonyl region in the DPPC bilayer [30,31]. Whereas the midpoint of the broad C=O stretching band shifts by ~2 cm^−1^ to higher frequencies at the pretransition temperature, at the main transition there is a shift of ~4 cm^−1^ back to lower frequencies [30]. The conjugates of 1-ANISA-2-PA-PC and 1-PA-2-ANISA-PC caused a slight shift in the vibration of C=O groups toward lower wavenumber values for the hydrated bilayer at 25 °C and 38 °C (Table 1). This may indicate changes in the degree of hydration of the interfacial region of the lipid bilayer. For the dehydrated film of conjugates/DPPC system, there is no temperature dependence of the C=O vibration, but 1-ANISA-2-PA-PC and 1-PA-2-ANISA-PC cause a shift in the vibration maximum toward lower wavenumber values (Appendix A). The lipid C=O band most likely overlaps with C=O groups originating from anisic acid moieties (1682 cm^−1^).

The scissoring mode is represented by the band at 1468 cm^−1^ (Figure 5a). All changes in the region that are in line with this mode are used to diagnose both the lipid hydrocarbon phase transition and alkyl chain packing arrangement, due to the fact that they reflect concomitant increases in the hydrocarbon chain mobility and gauche rotamer proportion. The position of the δ(CH_2_) vibration is sensitive to the type of lateral alkyl chain packing. In the gel phase, this band is sharp and intensive, a fact that suggests hexagonal packing (Figure 5a). The intensity of the δ(CH_2_) band decreases and becomes broader with increasing temperature. The maximum of the band shifts in the direction of lower wavenumber (Table 1). The presence of compounds 1-ANISA-2-PA-PC and 1-PA-2-ANISA-PC in the bilayer of liposomes shifts from the δ(CH_2_) vibrational band of the group towards lower wave number values (Table 1), which indicates that these compounds decrease in the organization of the membrane structure in the intermediate and liquid phases.

The phosphate moiety of the head group gave rise to several vibrations in the infrared. Asymmetric and symmetric stretching vibrations for the PO_2_^−^ group were found in the range 1255−1225 cm^−1^ and 1095−1060 cm^−1^, respectively (Table 1 and Appendix A). Moreover, in the 1370−1180 cm^−1^ region, there are series of bands that belong to the hydrocarbon chains. Nonetheless, it is difficult to observe this CH_2_ wagging band progression in the spectra of phospholipids because in the gel phase, it overlaps with the strong PO_2_—antisymmetric double stretching band. Heating is accompanied by the disappearance of CH_2_ wagging bands, indicating chain melting (Figure 7). 

The symmetric stretching mode at ~1087 cm^−1^ partially overlapped with the band, representing the C-O-P-O-C stretching modes ~1067 cm^−1^ (Figure 7). As shown in Figure 7, the bands of phosphate showed different behaviors with temperature. For example, in the region of the intermediate phase (temperature range T_p_–T_m_) and the liquid phase of DPPC bilayer, the presence of 1-PA-2-ANISA-PC caused a slight shift in the symmetric vibrational band (ν_s_(PO_2_^−^)) toward lower frequencies (Table 1), indicating an interaction through hydrogen bonds between the PO_2_^−^ group in lipids and anisic acid molecules. The frequencies of asymmetric vibration *v*_as_(PO_2_^−^) for phosphate are sensitive to the state of hydration of phospholipids bilayers. Dehydration results in band-shifts towards higher wavenumber. In the liquid crystal phase of the bilayer at a temperature of about 50 °C, the presence of 1-ANISA-2-PA-PC and 1-PA-2-ANISA-PC reduced the wavenumber of the PO_2_^−^ asymmetric band in the lipid membrane, which indicates that the degree of hydrogen bonding to the PO_2_^−^ group increased. Interestingly, in the temperature range of 36–38 °C for conjugates of liposomes with *p*-anisic acid (1-ANISA-2-PA-PC), the wavenumbers were shifted towards higher values compared to the control which, in turn, may indicate a reduction in hydration in this area. This is in line with the results from DSC, in which pretransition was eliminated with a molar ratio of the conjugate 1-ANISA-2-PA-PC:DPPC 1:5. It is a well-known fact that the type and number of thermotropic phases strongly depend on the degree of hydration in the lipid bilayer.

## 3. Summary

In this work we present a new form of liposomes composed of DPPC and conjugates of anisic acid with phosphatidylcholine. The active substance was incorporated into the sn-1 or sn-2 position of phospholipids. Moreover, the most important part of this study was its efforts to understand the impact of these conjugates (1-ANISA-2-PA-PC and 1-PA-2-ANISA-PC) on the physico-chemical properties of a liposome’s membrane. The data obtained from these studies are important in order to design such carriers that will allow for the better release of the incorporated substance. Arouri and Mouritsen [18] found that the premixing of conjugates with phospholipids can be used to modify physicochemical properties of liposomal formulations, which may be useful for further exploration into their potential in anti-cancer treatments.

It is also important that the secretory phospholipases A2 (sPLA2) is only active toward lipid aggregates, and the enzymatic activity is controlled by the lipid composition, morphology and physicochemical properties of the liposome membrane. Furthermore, the activity of sPLA2 enzymes is highly controlled by the phase of the lipid bilayer [32] and reaches maximum activity during the phase transition. As it is necessary to determine the temperature of the phase transition due to the fact that it may enhance the sPLA2 enzymatic activity on the lipid bilayer [32], it is therefore very important to accurately characterize the lipid membrane in order to predict and control enzyme activity.

All in all, our research shows that the new bioconjugates 1-ANISA-2-PA-PC and 1-PA-2-ANISA-PC, have a significant impact on the physico-chemical properties of the membrane and cause a decrease up to about two degrees (for 1-ANISA-2-PA-PC) in the temperature of the main phase transition, an observation which was confirmed using all previously mentioned methods. One consequence of this is the greater fluidity of the lipid bilayer. According to our knowledge, there have been no studies in which the impact of new carriers on the physico-chemical properties of the liposome membrane was determined to such a wide extent.

## 4. Materials and Methods

### 4.1. Chemicals and Reagents

*p*-anisic acid (ANISA) was coupled to *sn*-glycerol-3-phosphocholine at *sn*-1 or *sn*-2 position and two conjugates were obtained: 1-anisoyl-2-palmitoyl-*sn*-glycero-3-phosphocholine (1-ANISA-2-PA-PC) and 1-palmitoyl-2-anisoyl-*sn*-glycero-3-phosphocholine (1-PA-2-ANISA-PC). The compounds were originally synthesized at the Department of Chemistry, Wrocław University of Environmental and Life Sciences as described previously [12]. The structures of the compounds are shown in Figure 1. Three fluorescent probes, Merocyanine 540 (MC540), 6-dodecanoyl-2-dimethylaminonaphthalene (Laurdan) and 1,6-diphenyl-1,3,5-hexatriene (DPH), were purchased from Molecular Probes (Eugene, OR, USA). Lipids 1,2-dipalmitoyl-*sn*-glycero-3-phosphatidylcholine (DPPC, purity 99 %) and sterile filtered water were purchased from Sigma-Aldrich (St. Louis, MO, USA). Chloroform was purchased from POCH (Gliwice, Poland).

### 4.2. Liposome Preparation

For our research both multilamellar (MLVs) and small unilamellar vesicles (SUVs), which were formed from DPPC and the conjugates: 1-ANISA-2-PA-PC or 1-PA-2-ANISA-PC or ANISA, were used. Liposomes were formed in the specified molar ratio of the conjugates/DPPC: 1:5; 1:10; 1:25 and 1:50. We chose DPPC to formulation due to the fact that it is broadly described in the literature. Moreover, its phase transition temperature is at about 42 °C, which facilitates the formation of liposomes that have a transitional region close to human body temperature [33]. DPPC and the conjugates of phosphatidylcholine were dissolved in chloroform. Then, solvent was carefully evaporated to dryness under nitrogen, and a thin lipid film was formed on the flask wall. Samples were left in a vacuum pump for at least 2 h and then 3 mL of distilled water was added, and the lipid film was washed away from the flask wall using a mechanical shaker. Shaking was conducted at a temperature above the main phase transition of lipids, until all lipids created a homogeneous milky suspension of multilamellar vesicles (MLVs). Small unilamellar liposomes (SUVs) with a diameter of 100–120 nm were obtained by sonicating MLVs using a Sonics VCX750 sonicator (Sonics, Newtown, CT, USA) for 15 min at 20 kHz. The measurement was performed using Zetasizer Nano ZS (Malvern Panalytical, UK). MLVs were used in the calorimetric and FTIR studies and SUVs in the fluorimetric method.

### 4.3. Differential Scanning Calorimetry (DSC)

This method was described precisely in our previous paper [27] and it was used in order to follow the thermotropic behavior of MLVs prepared from DPPC and 1-ANISA-2-PA-PC, 1-PA-2-ANISA-PC or ANISA. For this study the molar ratios of conjugate to DPPC used were 1:5; 1:10; 1:25 and 1:50. The concentration of the lipid in the sample was 25 mg/mL. Preparation of liposomes was described in detail in Section 4.2. Liposomes were placed in Mettler Toledo standard aluminum crucibles of 40 μL capacity. Tightly closed vessels were incubated for 1 and 7 days at 4 °C. Measurements were performed using Mettler Toledo Thermal Analysis System D.S.C. 821^e^ (Mettler Toledo, LLC, Columbus, OH, USA) operated at a heating rate of 2 °C min^−1^ from 25 to 55 °C. Thermal cycles were repeated three times. Data analysis was performed using original software provided by Mettler Toledo in order to determine the temperature of the pre- (T_p_) and main transition (T_m_), the main half-width transition (ΔT_1/2_) and the calorimetric enthalpy (ΔH).

### 4.4. Spectroscopy Methods

In order to extend the research on the influence of conjugates on the thermotropic parameters of liposomes, spectroscopic methods, i.e., steady-state fluorescence and ATR-FTIR, were also used.

#### 4.4.1. Steady-State Fluorescence Spectroscopy

The fluorimetric method was described in our previous work [27,34]. In this research SUVs at the 1:5 molar ratio of conjugate/DPPC were used. The concentration of the lipid in the sample was 0.1 mg/mL. Fluorimetric studies were performed using three fluorescent probes, namely Laurdan, DPH and MC540, located in different areas of the membrane. Laurdan was located in hydrophilic–hydrophobic regions in the membrane. MC540 was slightly higher than Laurdan. Finally, DPH was located in a hydrophobic area. The fluorescent probes were added to the samples at concentration was 1 µM and after that, samples were incubated for 30 min in darkness at room temperature. Control samples contained only liposomes formed from DPPC with fluorescent probes (the molar ratios of probe:DPPC was 1:7000). Measurements were conducted above and below the main phase transition of DPPC in a range from 25 to 55 °C using CARY Eclipse fluorimeter (Varian, San Diego, CA, USA) equipped with DBS Peltier temperature controller (temp. accuracy ± 0.1 °C). The experiment was repeated three times.

Based on fluorescence intensity of Laurdan probe, general polarization (GP) was calculated using the formula:GP = (I_g_ − I_lc_)/(I_g_ + I_lc_),(1)
where, I_g_ and I_lc_—fluorescence intensities at gel (440 nm) and at fluid (490 nm) phase, respectively [24,25].

The DPH results were presented as changes in fluorescence anisotropy (A) and were calculated using the formula:A = (I_‖_ − GI_⊥_)/(I_‖_ + 2GI_⊥_),(2)
where, I_‖_ and I_⊥_ are fluorescence intensities observed in directions parallel and perpendicular to the polarization direction of the exciting wave, respectively. *G* is an apparatus constant dependent on the emission of wavelength [35]. The excitation and emission wavelengths for DPH were 360 nm and 425 nm; for MC540, they were 540 nm and 585 nm, respectively. The excitation wavelength for Laurdan was 360 nm, and the emitted fluorescence was recorded at two wavelengths: 440 and 490 nm.

#### 4.4.2. Attenuated Total Reflectance Fourier Transform Infrared Spectroscopy (ATR-FTIR)

ATR-FTIR is one of the most powerful methods for recording IR spectra of biological materials in general, and for biological membranes in particular. This method was based on the method described in our previous manuscript [27,36] with minor modifications. The concentration of the lipid in the sample was 0.1 mg/mL. The MLVs were prepared at the molar ratio of 1:5 and were placed on the ZnSe crystal (with thermostat). Next, the dissolvent was evaporated, and the spectrum of dry lipids was recorded in the range of temperature from 25 to 55 °C. After that, the lipids’ dry film was hydrated in the sterile filtered water solution, and the measurement was repeated. Measurements were performed using Thermo Nicolet 6700 MCT spectroscope (Thermo Fisher Scientific, Waltham, MA, USA). Each single spectrum was obtained from 128 records at 2 cm^−1^ resolution in the range from 700–4000 cm^−1^. After filtering, the noise out from the spectrum of the object was studied, the spectrum of the water was subtracted in order to remove a strong band of water and the baseline was corrected. The elaboration of a spectrum was carried out using EZ OMNIC v 8.0 program and Thermo Nicolet as well. The overlapping bands were deconvoluted and derivatized with commercial EZ OMNIC v 8.0 software using a sum of Gaussian and Lorentzian functions.

In the spectrum of liposomes formed from DPPC and the conjugates of phosphatidylcholine with p-anisic acid, we considered a thermotropic behavior in characteristic regions, namely the hydrocarbon chains, the interface region (carbonyl) and the head group (including choline and phosphate bands).

## 5. Conclusions

The aim of this research was to conduct a detailed analysis of the physico-chemical properties of new liposomes assembled from the bioconjugates of anisic acid with phosphatidylcholine in a wide range of temperatures. Using complementary methods, i.e., DSC, IR and fluorimetry, it was found that the tested compounds incorporated into the lipid bilayer are responsible for a decrease in the temperature of the main phase transition (particularly 1-ANISA-2-PA-PC), a slight increase in the fluidity of the membrane, and a change in hydration. They also have an impact on the dynamics and structure of the lipid bilayer. At the same time, they do not change the stability of liposomes in a significant way. Studies focused on physico-chemical properties are necessary for a thorough analysis of the stability and properties of new liposomes and constitute the basis for further research on the biological activity and pharmacological effectiveness of these substances as potential anticancer drugs.

## Figures and Tables

**Figure 1 ijms-22-13146-f001:**
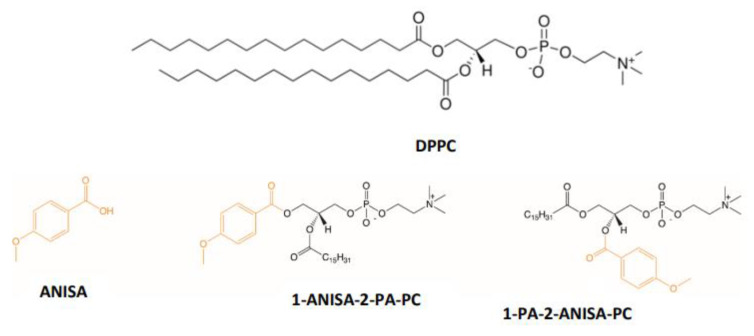
Structures of investigated *p*-anisic acid (ANISA) and its bioconjugates with phosphatidylcholine: 1-anisoyl-2-palmitoyl-*sn*-glycero-3-phosphocholine (1-ANISA-2-PA-PC) and 1-palmitoyl-2-anisoyl-*sn*-glycero-3-phosphocholine (1-PA-2-ANISA-PC) and DPPC.

**Figure 2 ijms-22-13146-f002:**
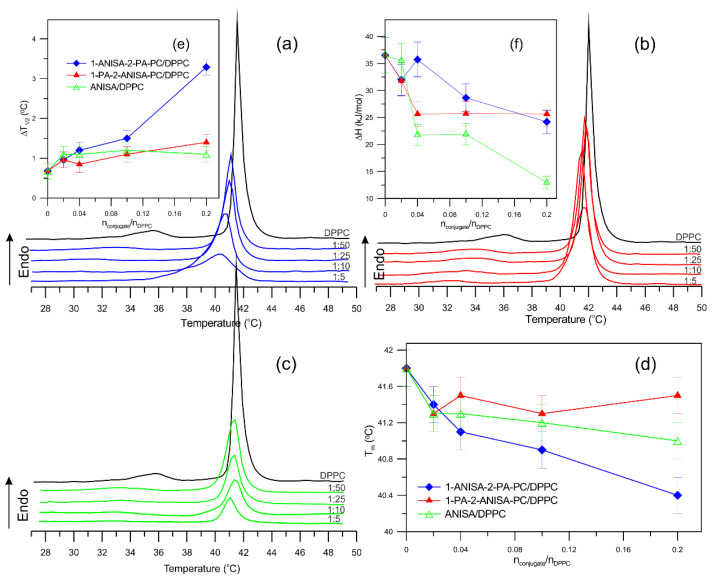
DSC thermogram of DPPC liposomes and new form of liposomes composed of DPPC and conjugates of anisic acid with phosphatidylcholine: (**a**) DPPC with 1-ANISA-2-PA-PC, (**b**) DPPC with 1-PA-2-ANISA-PC, and (**c**) DPPC with ANISA; (**d**) Main phase transition temperatures (T_m_) as a function of concentration; (**e**) Half-width peak (ΔT_1/2_) as a function of concentration; (**f**) Enthalpy change (∆H) of the main phase transition as a function concentration ofcompounds. The conjugate:DPPC molar ratios 1:5; 1:10; 1:25; 1:50.

**Figure 3 ijms-22-13146-f003:**
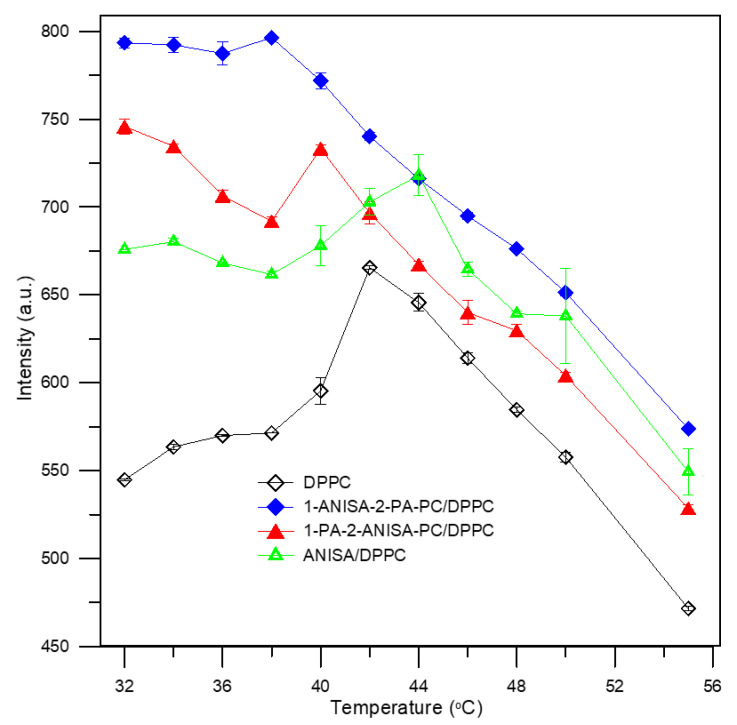
Fluorescence intensity of MC540 probe measured at 585 nm for liposomes formed from DPPC (control) and new forms of liposomes (1-ANISA-2-PA-PC/DPPC, 1-PA-2-ANISA-PC/DPPC and ANISA/DPPC) (tested probes) at molar ratio of 1:5, as a function of temperature.

**Figure 4 ijms-22-13146-f004:**
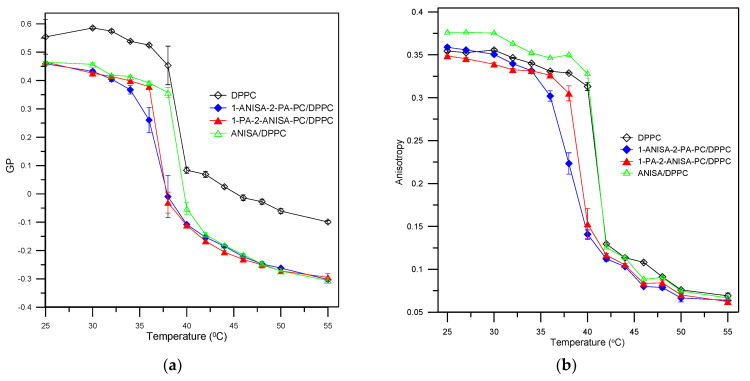
Values of general polarization (GP) of Laurdan in liposomes DPPC (control) and new forms of liposomes (tested probes) at molar ratio the conjugate:DPPC 1:5, as a function of temperature (**a**). Values of anisotropy of DPH in liposomes DPPC (control) and new forms of liposomes (tested probes) at molar ratio the conjugate:DPPC 1:5, as a function of temperature (**b**).

**Figure 5 ijms-22-13146-f005:**
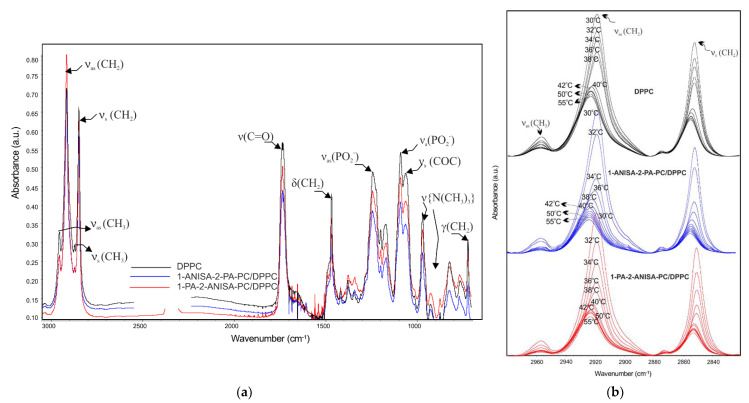
ATR-IR spectra of dehydrated DPPC film (control) and dehydrated conjugates/DPPC film at 25 °C with the assignment of the main absorption bands of DPPC. (**a**) Selected ATR-IR spectra of hydrated DPPC bilayer (control) and hydrated conjugates/DPPC bilayer in the hydrocarbon chains region at different temperatures. (**b**) The conjugate:DPPC molar ratio of 1:5.

**Figure 6 ijms-22-13146-f006:**
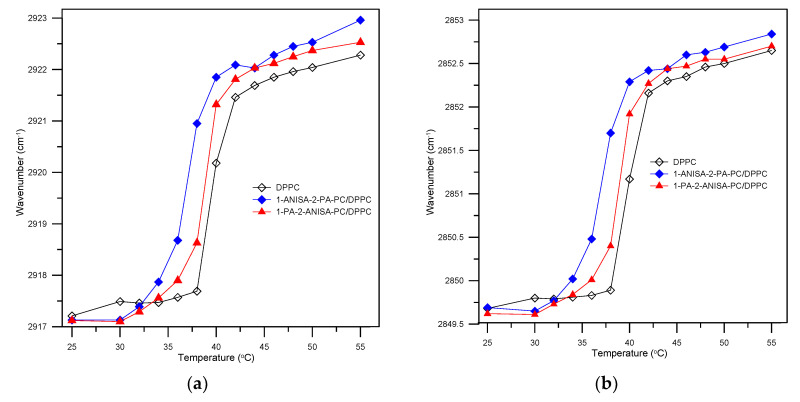
Temperature dependence of the asymmetric (**a**) and symmetric (**b**) CH_2_ stretching mode in the presence and absence of conjugates of anisic acid with phospatidylcholine (1-ANISA-2-PA-PC and 1-PA-2-ANISA-PC) for DPPC bilayers.

**Figure 7 ijms-22-13146-f007:**
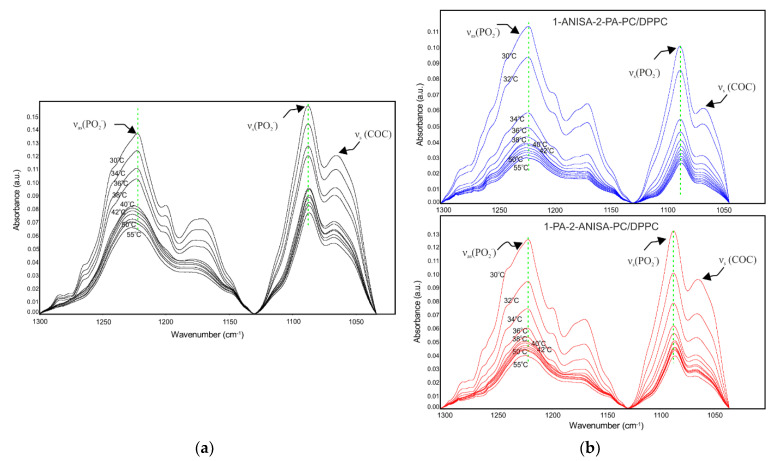
ATR-IR spectra of hydrated DPPC bilayer (**a**) and hydrated 1−ANISA−2−PA−PC/DPPC and 1−PA−2−ANISA−PC/DPPC bilayer (**b**) in the region between 1300–1040 cm^−1^ at different temperatures; the conjugate:DPPC molar ratio of 1:5.

**Table 1 ijms-22-13146-t001:** Assigned bands of ATR-FTIR spectra of hydrated DPPC bilayer and of hydrated 1-ANISA-2-PA-PC/DPPC (1:5) and 1-PA-2-ANISA-PC/DPPC (1:5) bilayer at different temperatures. The conjugate:DPPC molar ratio was 1:5.

ATR Spectra, Wavenumbers (cm^−1^)			
Assigned *	DPPC	1-ANISA-2-PA-PC/DPPC	1-PA-2-ANISA-PC/DPPC
25 °C			
δ(CH_2_)	1467.66	1467.53	1467.37
γ(CH_2_)	720.44	720.88	721.06
ν_s_(N-C)	925.39	925.39	926.07
ν_as_(N-C)_ip_	970.31	970.93	969.85
ν_s_(COP)	1064.92	1066.32	1064.65
ν_s_(PO_2_^−^)	1087.80	1088.40	1088.94
ν_as_(PO_2_^−^)	1221.61	1221.47	1221.82
ν(C=O)	1733.94	1733.52	1733.74
ν_s_(CH_2_)	2849.71	2849.69	2849.62
ν_s_(CH_3_)	2872.91	2872.75	2872.83
ν_as_(CH_2_)	2917.26	2917.13	2917.12
ν_as_(CH_3_)	2955.57	2955.53	2955.53
38 °C			
δ(CH_2_)	1467.73	1467.63	1467.69
γ_r_(CH_2_)	720.44	721.08	721.06
ν_s_(N-C)	925.39	925.95	926.07
ν_as_(N-C)_ip_	969.61	971.90	969.85
ν_s_(COP)	1066.61	1068.58	1068.44
ν_s_(PO_2_^−^)	1087.69	1087.49	1087.24
ν_as_(PO_2_^−^)	1221.86	1225.89	1221.27
ν(C=O)	1736.55	1735.46	1733.50
ν_s_(CH_2_)	2849.87	2851.71	2850.40
ν_s_(CH_3_)	2872.73	2872.63	2872.68
ν_as_(CH_2_)	2917.71	2920.99	2918.63
ν_as_(CH_3_)	2956.57	2956.53	2956.31
50 °C			
δ(CH_2_)	1465.32	1464.78	1464.88
γ_r_(CH_2_)	ND	ND	ND
ν_s_(N-C)	ND	ND	ND
ν_as_ N-C)_ip_	967.77	971.81	969.85
ν_s_(COP)	1068.22	1069.32	1068.22
ν_s_(PO_2_^−^)	1087.02	1086.95	1086.22
ν_as_(PO_2_^−^)	1220.47	1215.09	1215.48
ν(C=O)	1731.56	ND	ND
ν_s_(CH_2_)	2852.74	2852.82	2852.70
ν_s_(CH_3_)	ND	ND	ND
ν_as_(CH_2_)	2922.06	2922.74	2922.53
ν_as_(CH_3_)	2956.56	2956.32	2956.45

* vibrations: δ—bending; γ—deformation; ν—stretching; r—rocking; s—symmetric; as—antisymmetric; ND—not detected.

## Data Availability

The data presented in this study are available on request from the corresponding authors.

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
