# Peer review of "Evaluation of the Physico-Chemical Properties of Liposomes Assembled from Bioconjugates of Anisic Acid with Phosphatidylcholine"

_ijms, 2021, doi:10.3390/ijms222313146_

Round 1

Reviewer 1 Report

In this manuscript Pruchnik and co-workers reported on the physico-chemical properties of liposomes assembled from bioconjugates of anisic acid with phosphatidylcholine. To characterize those properties the Authors applied different methods and the obtained results are interesting and important addressing then problem of formulations of improved  mechanism of drug delivery systems. However, before my recommendation for publication  some problems and question are addressed.

The Authors have previously characterized similar systems as given in references 10-14. Since the number of possible modification is very big it would be necessary to clearly indicate relevance of this work.  What does this work adds to knowledge of previously described effects.

In general the experiments are well done and described. But there is also one general lack of information in Materials and Methods on quantitative data regarding concentration of used chemicals during experiments, except for fluorophores, at least such information should be given at Figure caption. For DSC measurements usually information about concentration of liposomes is given in mg/ml. The experiments were carried out on the freshly prepared sample or seasoned for hours or days? For both methods of liposomes preparations the sizes from DLS measurements should be given. Also for fluorescence measurements the liposome concentrations are usually lower to avoid scattering. Any way in each applied method the concentrations of the used chemicals should be added.

Another problem should be explained is connected with solubilization  of modified liposomes  in DPPC especially at higher concentrations because of possible formation of different aggregates.

Line 253  no Table S1  was given in Supplementary file

Figure 5b is hard to follow, maybe the results should be presented as a plot of peak position versus temperature

Lines 484-485 -  how this dry lipid was hydrated, immersed in water or added some drops of water, how you determined that is hydrated,

Discussion part is very short and taking into consideration the fact that lines 363-374 are rather belong to Introduction and repeat the information given therein. It seems to me that most of discussion is given after results presentation so maybe The Authors should change chapter Results to Results and Discussion.

Finally, the manuscript will be suitable for publication after mentioned above corrections and  improvements.

Author Response

Dear Reviewer,

We would like to thank you for your constructive comments and suggestions which are extremely valuable in order to improve the quality of our paper. Following the Reviewer’s suggestions, we revised the manuscript and made certain changes in the text which are highlighted in yellow.

The responses for your questions and suggestions are in enclosed file.

With kind regards

Aleksandra WÅ‚och

Reviewer 2 Report

The manuscript entitled  “Evaluation of the Physico-Chemical Properties of Liposomes Assembled from Bioconjugates of Anisic Acid with Phosphatidylcholine”  reports an  interesting examination of the effect of bioconjugates of anisic acid with phosphatidylcholine molecules on the thermodynamic and structural properties of DPPC membranes. The researchers employed many different experimental methods with rather carefully performed measurements.  This work is valuable and deserves a publication, however several points should be checked or addressed:

  1. Materials and Methods: Preparation of DPPC liposomes - it is not defined what is the final concentration of lipids in water dispersion of liposomes. Authors used a different methods, which need a different liposome concentration, there is any information about that. What is the molar ratio between fluorescence probes and lipids?
  2. MC540 fluorescence probe: why the fluorescence intensity of this probe in the gel phase of pure DPPC increases, then reaches its maximum in Tm and finally decreases again? More detailed explanations about this are needed.
  3. GP of Laurdan is dependent not only on polarity/hydration changes in lipid membranes. Solvent relaxation, which changes a GP value, is also related to the packing of lipid membranes. Thus, the statement “A decrease in the values of GP means that more water molecules was incorporated into the hydrophilic–hydrophobic interface of the bilayer” cannot be always true. A decrease in the values of GP can be observed in the membrane where hydration is rather at the same level but lipid head-groups are less packed.  The measurement of Laurdan anisotropy is needed in this case to show more clearly the effect of hydration and lipid packing on GP values.
  4. Figure 4b, anisotropy of DPH: I’m not really agree with the statement “The results indicated that conjugates of 1-ANISA-2-PA-PC and 1-PA-2-ANISA-PC 225 decrease the value of anisotropy in the bilayer in the gel and liquid phase compared to the control.” In my opinion the effect is only on Tm but not on anisotropy values present in both phases, they are pretty similar.
  5. “…1-ANISA-2-PA-PC and 1-PA-2-ANISA-PC cause a shift in the vibration maximum toward lower wavenumbers values (Table S1). This probably indicates changes in the degree of hydration of the interfacial region of the lipid bilayer.” If this is really dry lipid film, the hydration of lipid membranes shouldn’t change during experiment. I think most probably lipid niC=O band overlaps with niC=O groups originating from anisic acid moieties.  
  6. Table S1 is not present in supplementary materials.
  7. In the gel lipid phase the progression of a few wCH bands strongly overlaps with the nasPO2- band, so how did the authors determine the maximum of the nasPO2- band for the gel phase?

Author Response

(The authors gave the same response as above.)

Round 2

Reviewer 2 Report

accept for publication